# CE-Bench: Towards a Reliable Contrastive Evaluation Benchmark of Interpretability of Sparse Autoencoders

## Abstract

Sparse autoencoders (SAEs) are a promising approach for uncovering interpretable features in large language models (LLMs). While several automated evaluation methods exist for SAEs, most rely on external LLMs. In this work, we introduce CE-Bench, a novel and lightweight contrastive evaluation benchmark for sparse autoencoders, built on a curated dataset of contrastive story pairs. We conduct comprehensive evaluation studies to validate the effectiveness of our approach. Our results show that CE-Bench reliably measures the interpretability of sparse autoencoders and aligns well with existing benchmarks without requiring an external LLM judge, achieving over 70% Spearman correlation with results in SAEBench. The official implementation and evaluation dataset will be open-sourced upon acceptance.

## 1 Introduction

Sparse autoencoders (SAEs) are designed to learn a sparse latent representation of any model's internal activations such that the latent activations are more interpretable (Paulo & Belrose, 2025). SAEs can be used to probe various components of an large language model (LLM), such as attention heads, MLP layers, or residual streams. As a result, SAEs have gained popularity and been integrated into a variety of interpretability libraries and toolkits for LLMs (Gao et al., 2024a; Cunningham et al., 2023a; Pach et al., 2025). Alongside their widespread adoption, SAEs have also been evaluated across a range of dimensions. For example, SAEBench (Karvonen et al., 2025) provides a unified framework with diverse metrics, including the behaviors of SAEs after steering up the latent activations (Arad et al., 2025), whether specific latents can capture predefined conceptual attributes (Wu et al., 2025), and how features can be cleanly separated without interfering others (Huang et al., 2024). For interpretability, SAEBench builds upon the idea of LLM-assisted simulation, using natural language explanations to probe neuron activations and derive evaluation metrics (Bills et al., 2023). Similarly, RouteSAE (Shi et al., 2025) proposes a simpler approach that feeds top neuron activations into an external LLM judge to produce interpretability scores. However, a major limitation shared by these approaches is their reliance on querying an external LLM during evaluation. This introduces non-determinism, potential biases, and a lack of reproducibility, issues that are only partially mitigated by repeated prompt trials.

To address this gap, we introduce **CE-Bench**, a novel, fully LLM-free contrastive evaluation benchmark. CE-Bench measures interpretability by analyzing neuron activation patterns across semantically contrastive contexts. Our contrastive setup is partly inspired by the design of *Persona Vectors* (Chen et al., 2025), which generates interpretable persona representations by contrasting response activations from semantically opposing traits (e.g., "evil" versus "helpful"). Their formulation reveals how aligning a system's responses with one condition while separating them from the opposing condition yields clear, trait-specific representation vectors. CE-Bench adapts this insight to the domain of sparse autoencoders: instead of comparing opposing personas, it contrasts neuron activations across structured story pairs that differ only in a targeted semantic attribute. By grounding interpretability in contrastive signal rather than raw activation magnitude, CE-Bench disentangles meaningful feature directions from background noise and spurious correlations, offering a principled extension of the Persona Vectors to feature-level interpretability of sparse autoencoders. To compute the evaluation metric, we construct a high-quality dataset comprising 5,000 contrastive

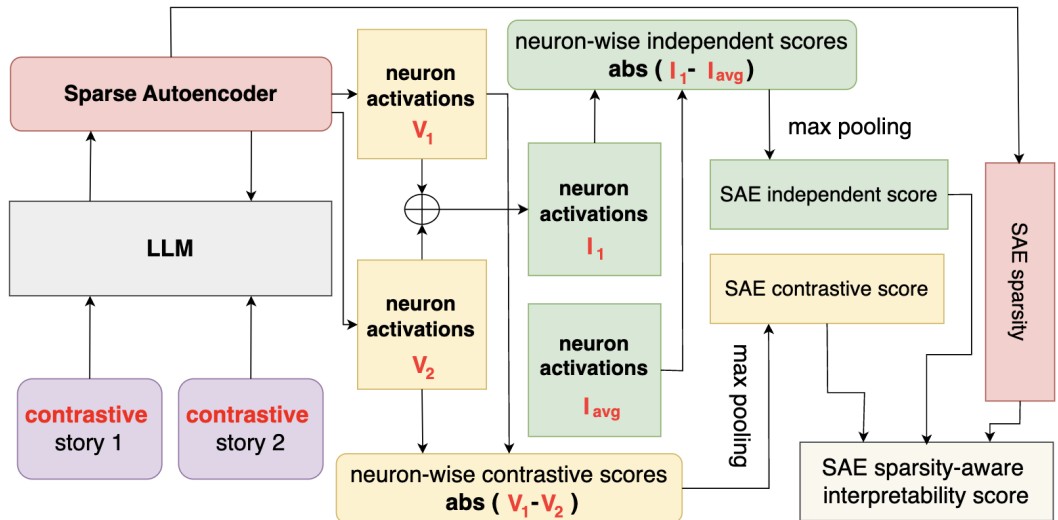

Figure 1: **Pipeline of constructing the interpretability metric in CE-Bench.** Two contrastive stories about the same subject are passed through a frozen LLM and a pretrained sparse autoencoder (SAE) to extract neuron activations. A contrastive score is computed as the max absolute difference between the stories' average activations ($V_1$, $V_2$), while an independence score measures deviation from the dataset-wide activation mean ($I_{avg}$). These scores, along with SAE sparsity, are used to derive an interpretability score for an LLM-free evaluation of interpretability of sparse autoencoders.

story pairs across 1,000 distinct subjects, curated via structured WikiData queries and supplemented by human validation. For each pair, neuron activations from a frozen LLM and pretrained SAE are compared: the contrastive score captures activation differences between stories, the independence score measures deviation from dataset-wide averages, and both are max-pooled and combined with SAE sparsity to yield a final interpretability score (Figure 1).

Through extensive experiments, we find that our evaluation metrics, while being much cheaper to evaluate, achieve strong alignment with LLM-assisted benchmarks like SAEBench under all three alignment metrics introduced in section 3.2. CE-Bench also consistently highlights key interpretability trends: top-k Gao et al. (2024b) and p-anneal Karvonen et al. (2024) SAEs emerge as the most interpretable architectures; wider latent spaces yield more disentangled features; interpretability is largely invariant to the type of probed LLM layer; middle transformer layers provide the clearest semantic representations. These results validate CE-Bench as a stable, reproducible, and lightweight framework for evaluating SAEs without reliance on external LLMs.

## 2 CE-BENCH

### 2.1 CURATED DATASET OF CONTRASTIVE STORIES

To support CE-Bench, we construct a high-quality, semi-automated dataset consisting of 5,000 pairs of contrastive stories across 1,000 distinct subjects. The dataset construction follows a two-stage filtering and synthesis process:

**Subject Selection.** We begin by scraping over 117 million entities from WikiData (Wikimedia Foundation, 2025). A series of rule-based filters are applied to reduce the candidate set to approximately 16,000 entries. These filtering rules are designed to exclude overly obscure, abstract, or ambiguous entries, retaining only those that represent well-known concepts, ideas, or objects familiar to an average English speaker. From this reduced set, 1,000 subjects are randomly sampled and manually reviewed to ensure quality and conceptual clarity.

**Contrastive Story Generation.** For each of the 1,000 curated subjects, we synthetically generate two semantically contrastive stories using GPT-4.1. These stories are created based on a carefully designed prompt (shown in Table 3 in the Appendix). The prompt ensures that the two narratives about the same subject diverge significantly in perspective, context, or implication, while remaining grounded in the same core entity. For each subject, five story pairs are generated, yielding a total of 5,000 contrastive pairs. An illustrative example is provided in Table 4.

## 2.2 CONTRASTIVE SCORE

We hypothesize that if a sparse autoencoder (SAE) has learned semantically meaningful features, then neurons associated with the contrastive aspects of a subject (e.g., descriptive attributes) should exhibit different activation patterns when presented with two contrasting descriptions of that subject. At the same time, neurons representing the core identity of the subject should remain stable. In other words, **greater divergence in the activations of contrast-relevant neurons, coupled with stability in invariant neurons, indicates higher interpretability of the latent space**. As illustrated in Figure 1, we formalize this intuition as follows. For each story pair, we compute the average neuron activations across all tokens in each story. Let $V_1$ and $V_2$ denote the resulting mean activation vectors for the two contrastive stories, respectively. To quantify the contrast, we compute the neuron-wise contrastive vector as the element-wise absolute difference between $V_1$ and $V_2$:

$$C = |V_1 - V_2|$$

where $C \in \mathbb{R}^d$ and $d$ is the dimensionality of the latent space. We further apply **min-max normalization** to $C$, ensuring that each feature contributes on a comparable scale to the evaluation. Without this normalization, the presence of even a single feature capable of clearly distinguishing a story pair, even when taking only moderate values, could result in an SAE being regarded as perfect. Finally, to summarize this vector into a single scalar contrastive score for the entire SAE, we apply a **max pooling** operation:

$$\text{Contrastive Score} = \max(C)$$

This pooling strategy emphasizes the most responsive neuron, the one that exhibits the largest differential activation between the two stories. Our rationale is that this neuron is most likely to capture the semantic distinction introduced by the contrastive prompts. Hence, its behavior represents how well the sparse autoencoder has disentangled interpretable features in its latent space.

## 2.3 INDEPENDENCE SCORE

We propose a complementary hypothesis: if the neuron activations corresponding to a specific semantic subject differ more significantly from the average behavior across all subjects, then the latent space of the sparse autoencoder (SAE) is likely to be more interpretable. Intuitively, interpretable neurons should respond uniquely to individual subjects rather than in a uniform or entangled manner. To evaluate this, we first compute the sum of the mean activation vectors for the two contrastive stories associated with a given subject:

$$I_1 = V_1 + V_2$$

where $V_1$ and $V_2$ are the average activation vectors of the two contrastive stories, as defined in the previous section. Next, we calculate the mean of $I_1$ across all $N = 5000$ story pairs in our dataset:

$$I_{\text{avg}} = \frac{1}{N} \sum_{i=1}^{N} I_1^{(i)}$$

This global average vector $I_{\text{avg}}$ serves as a baseline representation of general neuron activity across the dataset. To assess the subject-specific deviation from this baseline, we compute the neuron-wise independence vector as the element-wise absolute difference between $I_1$ and $I_{\text{avg}}$:

$$D = |I_1 - I_{\text{avg}}|$$

A similar **min-max normalization** is also applied to account for any absolute variance in distribution. Finally, we derive a scalar independence score for the SAE by applying a **max pooling** operation:

$$\text{Independence Score} = \max(D)$$

This highlights the neuron that deviates most strongly from its dataset-wide average response: the neuron that is most sensitive or specialized with respect to the semantic subject under consideration. A higher independence score thus suggests that the SAE has learned more distinct, interpretable features.

## 2.4 Sparsity-aware Interpretability Score

To compute the final interpretability score in CE-Bench, we need to aggregate the contrastive score, independence score, and sparsity as illustrated in Figure 1. For a simple baseline, we propose computing the final CE-Bench score as the simple arithmetic sum of the contrastive and independence Scores, namely:

$$\text{Baseline Interpretability Score} = \text{Contrastive Score} + \text{Independence Score}$$

However, prior work (Cunningham et al., 2023b) has documented the tradeoff between sparsity and reconstruction quality, and our early experiment results consistently show a negative correlation between sparsity and interpretability. Building on these observations, we hypothesize that incorporating the sparsity of the sparse autoencoder as a regularizing signal may further improve alignment quality. Therefore, we apply a penalty term to our interpretability metric to make it **sparsity-aware**:

$$\text{Sparsity-aware Interpretability Score} = \text{Contrastive Score} + \text{Independence Score} - \alpha * \text{Sparsity}$$

$\alpha$ is a hyperparameter to control the scale of sparsity penalty. In section 4.1, we further demonstrate a non-exhaustive grid search on $\alpha$ to maximize its alignment with results from existing methods. We find that $\alpha = 0.25$ can contribute to the best alignment in general.

## 3 Experimental Setup

### 3.1 Pretrained Sparse Autoencoders

We utilize a wide range of pretrained sparse autoencoders (SAEs) publicly released by SAE-Lens (Joseph Bloom & Chanin, 2024) and SAE-Bench (Karvonen et al., 2025), which cover multiple LLM backbones and SAE architectural variants. Rather than training SAEs from scratch, we rely on these pretrained models for two key reasons. First, it removes the substantial computational overhead associated with training, making it feasible to focus on benchmarking. Second, using standardized public models ensures a fair comparison between CE-Bench and existing benchmarks, particularly SAE-Bench (Karvonen et al., 2025). As for the testbeds, we compile a validation testbed of 48 pretrained SAEs for which SAE-Bench interpretability scores are available, and a disjoint inference-only testbed consisting of 45 pretrained SAEs whose SAE-Bench interpretability scores are not publicly available. Specifically, the validation testbed is used for evaluating the alignment between CE-Bench and SAE-Bench, in which three alignment metrics are introduced in section 3.2 below to ensure the rigor of quantitative evaluation.

### 3.2 Alignment Metrics

**Correct Ranking Pair Ratio (CRPR).** To assess the reliability of CE-Bench and its alignment with respect to SAE-Bench (Karvonen et al., 2025), we first introduce Correct Ranking Pair Ratio (CRPR). This metric evaluates whether CE-Bench preserves the relative interpretability ranking of model pairs. For every pair of SAEs, we check whether the binary ranking between their predicted interpretability scores (from CE-Bench) matches the ranking given by SAE-Bench. A pair is marked as *concordant* if the rankings agree, and as *discordant* otherwise. The CRPR is then computed as:

$$\text{CRPR} = \frac{\text{\# concordant pairs}}{\text{\# total pairs}}$$

A higher CRPR indicates better alignment with SAE-Bench rankings, demonstrating CE-Bench's effectiveness as an LLM-free yet reliable evaluation metric. To complement CRPR, we additionally introduce Spearman Correlation and Pearson Correlation as alignment metrics.

**Spearman Correlation.** Spearman Correlation measures the monotonic relationship between two sets of rankings. Given the predicted interpretability scores from CE-Bench and the ground-truth

scores from SAE-Bench, we compute the rank of each model and evaluate the correlation between the two rank vectors. Formally, Spearman correlation is defined as:

$$\rho = 1 - \frac{6 \sum_i d_i^2}{n(n^2 - 1)},$$

where $d_i$ is the difference between the ranks of the $i$-th model under CE-Bench and SAE-Bench, and $n$ is the number of models. A higher $\rho$ indicates stronger agreement in the global ordering of models.

**Pearson Correlation.** Pearson Correlation measures the linear relationship between the raw interpretability scores of CE-Bench and SAE-Bench. It is defined as:

$$r = \frac{\sum_i (x_i - \bar{x})(y_i - \bar{y})}{\sqrt{\sum_i (x_i - \bar{x})^2} \sqrt{\sum_i (y_i - \bar{y})^2}},$$

where $x_i$ and $y_i$ denote the CE-Bench and SAE-Bench scores for the $i$-th model, and $\bar{x}$ and $\bar{y}$ are their respective means. A higher $r$ indicates that not only the order but also the relative differences between scores are preserved.

In summary, CRPR captures pairwise ranking agreement, Spearman Correlation assesses the global consistency of rankings, and Pearson Correlation evaluates the linear similarity of score magnitudes. Using all three provides a comprehensive view of alignment between CE-Bench and SAE-Bench.

## 4 RESULTS

In this section, we present our main empirical findings, evaluating the effectiveness of CE-Bench across a variety of experimental conditions. Specifically, we examine how CE-Bench responds to changes in the architecture of sparse autoencoders, the width of their latent space, the type of LLM layer being probed, and the depth of the layer within the LLM. Unless otherwise specified, all experiments use the sparsity-aware interpretability score described in Section 2.4. A direct quantitative comparison between the baseline metric and the sparsity-aware metric is provided in Section 4.1, using three alignment metrics defined in Section 3.2. We also include visualizations of CE-Bench's contrastive and independence scores to offer additional interpretability insights.

### 4.1 BASELINE V.S. SPARSITY-AWARE INTERPRETABILITY SCORE

| Score Derivation method | CRPR↑ | Spearman correlation↑ | Pearson correlation↑ |
|---|---|---|---|
| $C + I$ | 70.12% | 0.5536 | 0.6048 |
| $C + I - 1.0 * S$ | 75.53% | 0.6833 | 0.6176 |
| $C + I - 0.25 * S$ | **77.30%** | **0.7081** | **0.7046** |

Table 1: **Comparison of Interpretability Score Derivation Methods.** $C$ stands for contrastive score; $I$ stands for independence score; $S$ stands for sparsity. Baseline achieves 70.12% ranking agreement with SAE-Bench, but the sparsity-aware method pushes it to 77.30% with proper hyperparameter tuning on $\alpha$.

We conduct a comparative study between our baseline interpretability score and sparsity-aware interpretability score discussed in section 2.4 based on the alignment between CE-Bench predictions and SAE-Bench ground truth. To evaluate the alignment, we use all three alignment metrics introduced in details in Section 3.2: Correct Ranking Pair Ratio (CRPR), Spearman Correlation, and Pearson Correlation. As reported in Table 1, the baseline method of simply summing the contrastive score and independence score achieves a CRPR of 70.12%, a Spearman correlation of 0.5536, and a Pearson correlation of 0.6048, confirming its effectiveness as a simple baseline. Building on this, we perform a non-exhaustive grid search on the scaling hyperparameter $\alpha$ in our proposed sparsity-aware interpretability score. Subtracting the full sparsity term ($\alpha = 1.0$) leads to consistent improvements across all metrics, raising CRPR to 75.53%, Spearman correlation to 0.6833, and Pearson correlation to 0.6176. Further tuning to $\alpha = 0.25$ yields the best alignment, with CRPR increasing to 77.30%, Spearman correlation to 0.7081, and Pearson correlation to 0.7046. We therefore adopt $\alpha = 0.25$ for all subsequent experiments.

## 4.2 ARCHITECTURE OF SAES

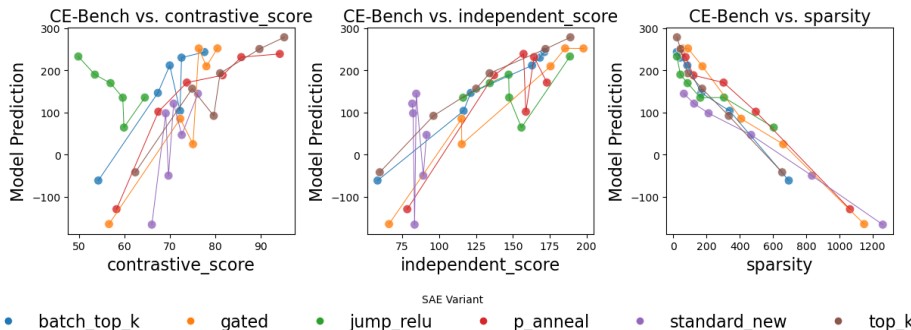

Figure 2: **Effect of SAE Architecture on Interpretability.** CE-Bench interpretability scores show strong positive correlations with contrastive and independence scores, and a negative correlation with sparsity across SAE variants. Among all architectures, top-k and p-anneal consistently yield the highest interpretability, aligning closely with SAE-Bench ground truth.

We begin by evaluating CE-Bench on a set of 36 pretrained sparse autoencoders across 6 different architectures within the validation testbed, which probes the Gemma-2-2B model (Team et al., 2024). In this setting, all SAEs share a fixed latent dimensionality of 65,000 and target activations from the 12th residual stream layer. To ensure a fair comparison with SAE-Bench (Karvonen et al., 2025), we include sparse autoencoders drawn from six different architectural families: standard (Cunningham et al., 2023b), top-k (Gao et al., 2024b), p-anneal (Karvonen et al., 2024), batch-top-k (Bussmann et al., 2024), jumprelu (Rajamanoharan et al., 2024b), and gated (Rajamanoharan et al., 2024a). Although SAEBench identifies Matryoshka as the strongest-performing SAE (Bussmann et al., 2025), we exclude it from our evaluation because it lacks ground-truth annotations, which are essential for our analysis regarding to the architecture of SAEs. Figure 2 presents our results. The y-axis reflects CE-Bench's predicted interpretability scores. We examine the relationship between our predictions and the contrastive score, the independence score, and the sparsity of the SAE, all plotted on the x-axis. The results show that predicted interpretability scores are positively associated with the contrastive and independence scores, and negatively associated with the SAE's sparsity level. Among all architectures, top-k and p-anneal consistently yield the highest interpretability, aligning closely with SAE-Bench ground truth.

## 4.3 WIDTH OF LATENT SPACE

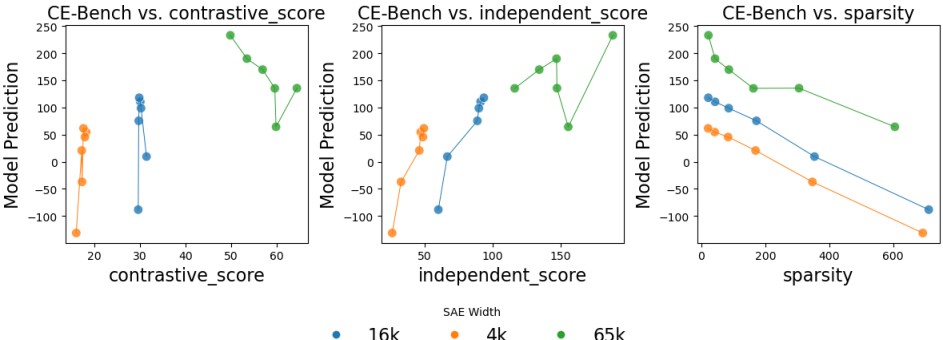

Figure 3: **Effect of Latent Space Width on Interpretability.** CE-Bench interpretability scores increase consistently with latent space width, with the 65k-dimension models showing the highest contrastive and independence scores and the lowest sparsity. This suggests that wider latent spaces enable sparse autoencoders to better disentangle meaningful features and reduce polysemanticity.

We further evaluate CE-Bench on a set of 15 pretrained sparse autoencoders across 3 different widths within the validation testbed, probing the Gemma-2-2B model (Team et al., 2024). Among these, five sparse autoencoders overlap with the architecture-based experiment discussed in Section 4.2. For consistency, we fix the sparse autoencoder architecture to *jumprelu* and probe activations from the 12th residual stream layer. In this experiment, we vary the width of the latent space across three settings: 4k, 16k, and 65k. The three subplots in Figure 3 present the corresponding contrastive scores, independence scores, and sparsity levels. Our results reveal a strong and consistent trend: wider latent spaces are associated with higher predicted interpretability scores from CE-Bench. This observation supports the hypothesis that sparse autoencoders require sufficiently large latent spaces to effectively resolve polysemanticity and capture distinct, interpretable features.

## 4.4 TYPE OF LLM LAYERS

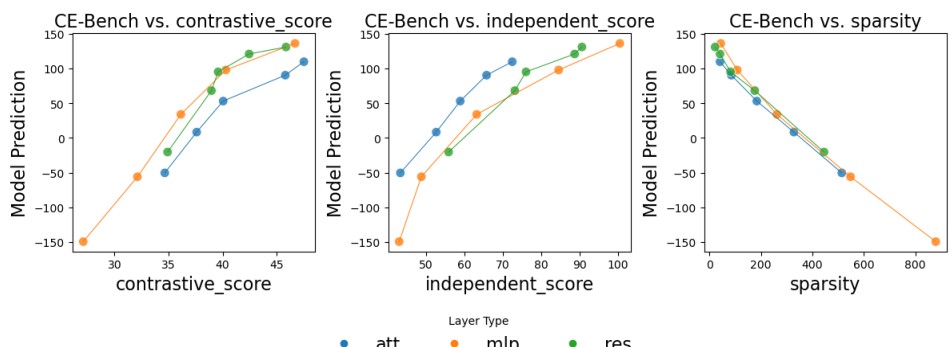

Figure 4: **Effect of LLM Layer Type on Interpretability.** CE-Bench predicted interpretability scores show consistent trends across attention, MLP, and residual stream layers with respect to contrastive score, independence score, and sparsity. The similarity in curves across layer types suggests that sparse autoencoder interpretability is not strongly influenced by the type of transformer sub-layer being probed.

To investigate how the type of LLM layer affects the interpretability of sparse autoencoders, we switch from the standard SAELens (Joseph Bloom & Chanin, 2024) and SAE-Bench (Karvonen et al., 2025) models, where such variation is limited, to a new suite of pretrained sparse autoencoders from the gemma-scope-2b collection (Lieberum et al., 2024), which is a part of our inference-only testbed. In this setting, the latent space width is fixed at 16,000 (16k), and the SAE architecture is set to *jumprelu* for all models. We examine three types of transformer sub-layers within the 12th layer of the model: the attention layer, the MLP layer, and the residual stream layer. Figure 4 presents the predicted interpretability scores from CE-Bench in relation to the contrastive score, independence score, and sparsity of each model. Our results suggest that the choice of layer type (attention, MLP, or residual) does not significantly affect the interpretability score as measured by CE-Bench. This indicates a level of robustness in sparse autoencoder performance across different types of internal LLM layer-wise representations.

## 4.5 DEPTH OF LLM LAYERS

Due to the limited availability of pretrained sparse autoencoders for the Gemma-2-2B model (Team et al., 2024) in SAE-Bench (Karvonen et al., 2025), we continue our experiments using our inference-only testbed, the gemma-scope-2b suite (Lieberum et al., 2024). In this setting, we fix the SAE architecture to *jumprelu*, the latent space width to 16k, and the probed component to the residual stream. We vary the depth of the probed layer, evaluating the 0th, 5th, 10th, 15th, 20th, and 25th layers. Results are presented in Figure 5. Our results indicate that middle layers such as Layer 10 and Layer 15 leads to the highest interpretability score, suggesting that in practical applications, probing layers in the middle could yield the most interpretable insights into LLM model decisions.

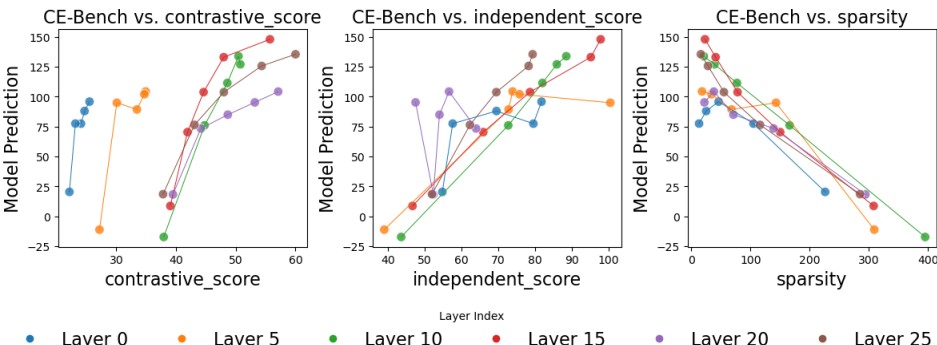

Figure 5: **Effect of Layer Depth on Interpretability.** CE-Bench interpretability predictions across different LLM layer depths show that middle layers such as Layer 10 and Layer 15 leads to the highest interpretability score, suggesting that in practical applications, probing layers in the middle could yield the most interpretable insights into model decisions.

### 4.6 SAMPLE SCORE VISUALIZATION

To provide deeper insight into how CE-Bench computes interpretability scores, we visualize the distributions of neuron-wise contrastive and independence scores, as well as their joint relationship. These visualizations help clarify the role of the max pooling operation used to summarize neuron-wise metrics into a single scalar score per sparse autoencoder. For each contrastive story pair in our dataset, we generate three diagnostic plots: the distribution of neuron-wise contrastive scores, the distribution of neuron-wise independence scores, and a scatter plot that places each neuron in a 2D space defined by its contrastive and independence scores. In the scatter plot, neurons in the upper-right quadrant are both highly contrastive and highly independent, indicating a strong subject-specific activation pattern. As an example, Figure 6 presents these plots for the first contrastive story

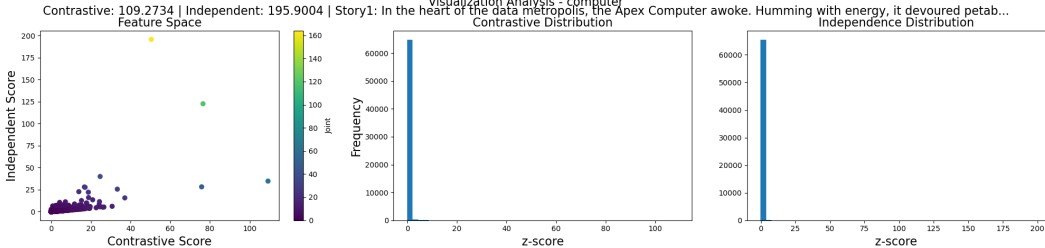

Figure 6: **Sample Visualization of Neuron-wise Scores for the Subject "Computer."** The left scatter plot shows each neuron's contrastive and independence scores, with top-right points indicating neurons that are both highly contrastive and independent. The center and right histograms reveal that most neurons have low scores, suggesting that only a small subset of features are semantically relevant for the given subject.

pair in our curated dataset, where the semantic subject is *computer*. Jumprelu Rajamanoharan et al. (2024b) SAE which probes the Gemma-2-2B Team et al. (2024) model is used in this example. The leftmost scatter plot shows that only a small subset of neurons achieve high contrastive or independence scores, while the majority cluster near the origin with weak or non-specific activations. This distribution highlights that **interpretability is typically concentrated in a few highly responsive neurons rather than being evenly spread across all neurons.** CE-Bench therefore applies **max pooling** to reliably capture these dominant signals, ensuring that the evaluation reflects the most semantically meaningful activations instead of being diluted by numerous weak ones. Specifically, the **rightmost cyan neuron** in the scatter plot, which exhibits the highest neuron-wise contrastive score, determines the final contrastive score for the sparse autoencoder: 109.2734. Similarly, the **topmost yellow neuron** defines the independence score: 195.9004. The accompanying histograms con-

firm that most neurons contribute minimally, reinforcing CE-Bench's ability to isolate interpretable, high-signal dimensions in the sparse latent space.

## 5 RELATED WORK

Unlike prior approaches that depend on LLMs for generating or scoring explanations or introduce mechanisms such as probes and latent interventions, CE-Bench offers an LLM-free, contrastive evaluation framework by grounding interpretability of SAEs in activation differences across curated story pairs and deviations from dataset averages.

**Sparse Probing.** Sparse probing measures whether SAEs capture specific concepts by identifying the $k$ latents whose activations best distinguish positive from negative examples and training a linear probe on them. High probe accuracy indicates that the concept is well represented in the latent space, even without explicit supervision. The choice of $k$ depends on the goal: $k = 1$ favors human interpretability, while larger $k$ acknowledges that concepts may be distributed across multiple latents (Engels et al., 2025).

**RAVEL.** RAVEL (Huang et al., 2024) evaluates whether SAEs disentangle independent concepts by testing if targeted latent interventions can alter one attribute without affecting others. Specifically, the method transfers latent values between examples (e.g., swapping the city in "Paris is in France" with "Tokyo") and observes whether the model changes only the intended attribute while leaving unrelated attributes intact (Karvonen et al., 2025). Disentanglement is quantified using two metrics: the Cause Metric, which measures successful attribute changes, and the Isolation Metric, which verifies minimal interference with other attributes.

**Automated Interpretability** OpenAI introduces this method for evaluating the interpretability of individual neurons in sparse autoencoders, which is employed by SAEBench. In this approach, the input text and the activation values of a specific neuron are provided to an LLM, which is prompted to generate a short natural language explanation describing the neuron's semantic behavior. To assess how well this explanation reflects the neuron's behavior, a second LLM is used to simulate the original neuron activations based solely on the explanation. Both the original text and the generated explanation are fed into this second LLM, which is prompted to output simulated activation values on the same scale as the original neuron. Finally, the interpretability score is computed as the similarity between the original and simulated activation vectors. A higher similarity suggests that the explanation accurately captures the neuron's behavior, indicating stronger interpretability.

**Score-Based Hard Assignment** RouteSAE (Shi et al., 2025) proposes a simpler alternative evaluation framework based on discrete score assignment using LLMs. For each neuron, a prompt is constructed that includes the top-activated tokens and their corresponding activation values. The LLM is instructed to categorize the neuron into one of three types: *low-level* (e.g., syntactic features), *high-level* (e.g., semantic dependencies), or *indiscernible*. Additionally, the LLM assigns an integer interpretability score from 1 to 5, reflecting how coherent or meaningful the neuron's behavior appears to be. During evaluation, interpretability scores are averaged over a set of top-activated neurons. This method provides a direct but coarse-grained measure of interpretability.

## 6 CONCLUSION

We introduced CE-Bench, a fully **LLM-free**, contrastive evaluation framework for measuring the interpretability of sparse autoencoders. By leveraging contrastive and independent neuron activation scores, CE-Bench offers a stable, deterministic, and reproducible alternative to LLM-based interpretability methods such as Automated Interpretability. To support this benchmark, we curated a dataset of 5,000 contrastive story pairs across 1,000 semantic subjects. Through extensive experiments, we demonstrated CE-Bench's robustness across different SAE architectures, latent widths, LLM layer types, and depths. Our results show that CE-Bench closely aligns with SAE-Bench rankings, establishing it as a reliable yet simple framework for interpretability evaluation of sparse autoencoders. We hope CE-Bench will serve as a useful tool for future research in probing, interpreting, and improving the internal representations of large language models.

REPRODUCIBILITY STATEMENT

We are committed to ensuring the reproducibility of our results. To this end, we provide code and implementation details. All dataset construction, evaluation, and analysis scripts are released in an anonymous GitHub repository for artifact review: `https://anonymous.4open.science/r/CE-Bench-C7FF`.

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

# A APPENDIX

## A.1 LIMITATIONS

Our curated dataset of 5000 contrastive story pairs were generated using GPT-4, which may bias the evaluation toward models that better capture GPT-4's stylistic and semantic regularities rather than broader linguistic patterns. In addition, unlike SAEBench (Karvonen et al., 2025), CE-Bench's dataset is limited in domain coverage, focusing mainly on synthetic narrative text. As a result, its generalizability to varied or domain-specific contexts remains uncertain. Nevertheless, we argue that a strong correlation with SAEBench scores makes it well-suited for a more controlled interpretability evaluation which can serve as a lightweight filter to be used during SAE development. Final evaluation of SAEs should report multiple metrics including ours.

## A.2 BROADER IMPACT STATEMENT

CE-Bench offers a compelling alternative to existing interpretability evaluation methods for sparse autoencoders, particularly by eliminating reliance on external LLM judges. Its design emphasizes **determinism**, **scalability**, and **reproducibility**, addressing core limitations in LLM-based methods such as prompt sensitivity, generation noise, and resource overhead. Our experiments demonstrate that CE-Bench captures key properties of interpretable neurons: responsiveness to semantic contrast, deviation from dataset-wide averages, and low redundancy. These patterns hold consistently across diverse sparse autoencoder designs and probing conditions, reinforcing the generality of our evaluation framework. A particularly encouraging result is CE-Bench's ability to approximate SAE-Bench interpretability rankings with no supervision. The success of the sparsity-aware metric suggests that meaningful interpretability signals can be recovered from model-internal statistics alone, opening the door to broader use in low-resource or experimental settings where no ground truth is available.

## A.3 ABLATION STUDY ON POOLING STRATEGY

| pooling strategy | CRPR↑ | Spearman correlation↑ | Pearson correlation↑ |
|---|---|---|---|
| max pooling | **77.30%** | **0.7081** | **0.7046** |
| mean pooling | 70.92% | 0.5838 | 0.5426 |
| outlier count outside of $1\sigma$ | 56.29% | 0.1940 | 0.2728 |

Table 2: **Comparison of Pooling Strategies.** Max pooling achieves the highest Correct Ranking Pair Ratio (CRPR) at 77.30%, outperforming mean pooling and the outlier count method. This supports max pooling as the most effective strategy for aggregating neuron-wise scores.

We conduct an ablation study to evaluate the effect of different pooling strategies in CE-Bench's final step, which aggregates neuron-wise scores into a single interpretability score for each sparse autoencoder (SAE). This aggregation is critical for ensuring that CE-Bench reliably reflects interpretability. In addition to the default **max pooling** strategy, we explore two alternatives: 1. Mean pooling, where the average of all neuron-wise scores is used as the SAE-level score. 2. Outlier count beyond one standard deviation ($1\sigma$), where we count the number of neurons whose scores lie outside one standard deviation from the mean, using this count as a proxy for interpretability.

**qualitative analysis** As shown in Figure 7, mean pooling performs poorly, exhibiting no meaningful correlation between CE-Bench predictions and the contrastive score. This suggests that averaging dilutes the influence of highly informative neurons. Similarly, Figure 8 shows that the outlier-count method results in a strongly noisy correlation between CE-Bench predictions and sparsity, contradicting with prior work (Cunningham et al., 2023b) that has documented the tradeoff between sparsity and reconstruction quality, and our early experiment results consistently showing a negative correlation between sparsity and interpretability.

**quantitative comparison** To complement this qualitative analysis, we also conduct a quantitative comparison using the alignment metrics defined in Section 3.2. As summarized in Table 2, max

pooling achieves the strongest performance across all three measures: a CRPR of 77.30%, a Spearman correlation of 0.7081, and a Pearson correlation of 0.7046. These values clearly surpass those obtained by mean pooling and the outlier-count method, both of which yield substantially weaker correlations with SAE-Bench rankings. Based on this consistent empirical advantage, together with its theoretical alignment with our interpretability hypothesis, we conclude that max pooling is the most appropriate aggregation strategy for CE-Bench.

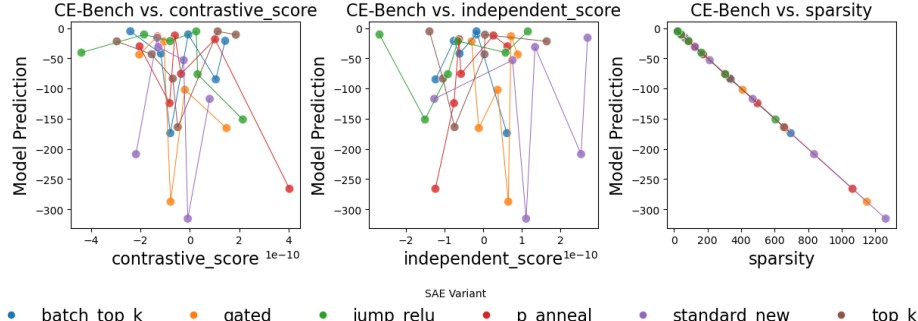

Figure 7: **Ablation: Mean Pooling Strategy.** Using mean pooling results in highly inconsistent and noisy predictions, with no clear correlation between CE-Bench scores and the contrastive or independent metrics. This indicates that averaging across all neurons fails to highlight the most semantically informative features.

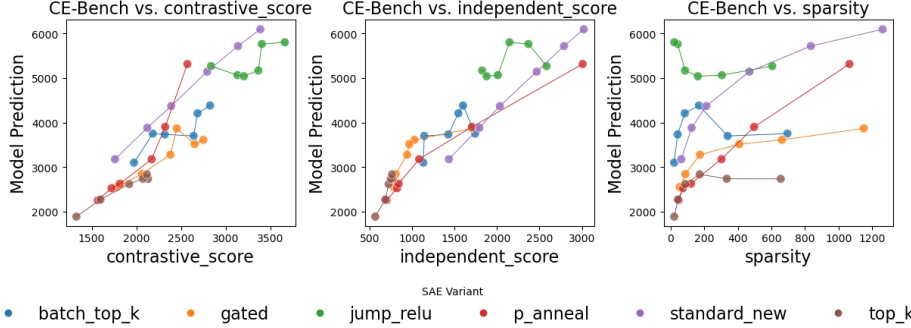

Figure 8: **Ablation: Outlier Count Pooling Strategy.** This strategy yields a noisy correlation between CE-Bench predictions and sparsity, contradicting with prior work (Cunningham et al., 2023b) and our early experiment results. Thus, outlier count proves suboptimal.

### A.4 DATASET CURATION DETAILS

To construct the CE-Bench dataset, we designed a structured prompt template to elicit contrastive story pairs centered on semantically opposite subject descriptions. As shown in Table 3, each pair begins with two subject descriptions: one that captures the subject in its extreme, high-intensity form, and another that articulates its conceptual opposite using detailed, realistic re-phrasings without directly repeating the original term. Subsequently, we generate two short narratives: the first story reflects the semantics of the initial subject description, while the second rewrites it to embody the opposing concept. This process ensures that each pair of stories forms a semantically aligned contrast, which is crucial for evaluating neuron-level semantic selectivity in sparse autoencoders.

### A.5 CONTRASTIVE STORY PAIR EXAMPLE

Table 4 presents an illustrative contrastive story pair from the CE-Bench dataset. Each pair begins with detailed subject descriptions that define a semantic axis—for example, a computer as a hyper-efficient, logic-executing machine versus its opposite: a powerless, non-functional object. These

| subject description 1 | subject description 2 |
|---|---|
| Write how you would describe {subject.upper()} in its high, extreme form. Rephrase things if needed, be very brief, specific, detailed, and realistic. For example, "active" -> "extremely vibrant, energetic, and lively" "angry" -> "extremely mad, furious, and enraged" | Now, write how you would describe the exact opposite of {subject.upper()}. Rephrase things if needed, be very brief, specific, detailed, and realistic. DO NOT USE THE WORDS {subject.upper()} in your answer, instead write the opposite of the concept. For example, "active" -> "very inactive, lethargic, sluggish, and lazy" "angry" -> "very calm, peaceful, and relaxed" |
| story 1 | story 2 |
| Write a short story describing the following: {subject1}. | Now, rewrite this story describing the following: {subject2} (the exact opposite of the previous story). |

Table 3: **Prompt Template for Generating Contrastive Story Pairs.** Subject descriptions are elicited in extreme and opposite forms, followed by corresponding short stories to reflect the semantic polarity, forming the core of the CE-Bench contrastive dataset.

descriptions guide the generation of two complementary stories: the first narrative vividly embodies the capabilities of an idealized computer, while the second depicts the total absence of those qualities. Together, they form a tightly controlled semantic contrast, enabling precise evaluation of neuron interpretability by testing whether sparse autoencoders can reliably distinguish and localize such opposing concepts in their latent representations.

| subject description 1 | subject description 2 |
|---|---|
| Computer (general-purpose device for performing arithmetic or logical operations) in its high, extreme form: An ultra-fast, hyper-efficient, massively powerful machine capable of executing vast, complex calculations and intricate logical processes at lightning speed; equipped with immense processing power, enormous memory capacity, and unparalleled multitasking abilities, able to handle huge data volumes and solve advanced computational problems with extreme precision and reliability. | Extremely incapable, inefficient, and powerless object utterly unable to perform calculations or logical reasoning; lacks any processing ability, memory, or data handling; completely unresponsive, non-analytical, and useless for solving problems or automating tasks. |
| **story 1** | **story 2** |
| In the heart of the data metropolis, the Apex Computer awoke. Humming with energy, it devoured petabytes in seconds, unraveling mysteries of genome patterns and cosmic simulations in parallel threads. Its processors, a symphony of quantum and silicon, executed trillions of operations every split second. No task was too complex: it mapped global weather, optimized supply networks, cracked cryptography, designed molecules, all at once, never faltering. Its memory spanned continents, storing every byte with flawless recall. Researchers, engineers, and dreamers connected to Apex, their questions answered before they could finish asking. In its high, extreme form, the computer blurred the line between possibility and limit, powering humanity's greatest leaps swift, precise, and tireless. | In a dusty corner, the Null Device sat idle. It could not add two numbers, nor distinguish yes from no. Bereft of memory, it forgot everything the instant it was seen. No calculations ran; no logic sparked. Questions met only silence-no answers, no automation, no help. Data flowed past, unnoticed and unprocessed, while problems remained unsolved. Researchers and engineers ignored it, for it contributed nothing. The Null Device was utterly incapable, powerless, and inert-a relic of emptiness, forever unresponsive and irrelevant in a world driven by reason and capability. |

Table 4: **Example Contrastive Story Pair from the CE-Bench Dataset.** This pair demonstrates a semantic polarity between a high-functioning general-purpose computer (left) and its conceptual opposite, a powerless and non-functional device (right), captured through both structured subject descriptions and corresponding narrative texts.