# OpenReview forum: "CE-Bench: Towards a Reliable Contrastive Evaluation Benchmark of Interpretability of Sparse Autoencoders"
_ICLR.cc/2026/Conference — ICLR 2026 Conference Withdrawn Submission_

### Official Review · Reviewer_8fSj · 2025-10-25

**Soundness:** 2
**Presentation:** 1
**Contribution:** 1
**Rating:** 0
**Confidence:** 5

**Summary:**

This paper proposes a benchmark for evaluating SAE quality that does not make use of an auxiliary LLM to judge the interpretability of latents. To construct the benchmark, the authors introduce a dataset of contrastive pairs of stories. They then collect sparse representations for these stories and use them to compute two metrics: contrastive score (measuring whether there is a single latent according to which stories in a contrastive pair are very different from each other) and independence score (measuring whether there is a single latent according to which the stories in a pair are different from the average over all stories). The overall quality metric combines these scores with a sparsity penalty; with appropriate hyperparameter tuning, this metric correlates moderately with the SAEBench quality metric introduced in prior work.

**Strengths:**

1. The idea of looking for individual latents that clearly separate contrastive pairs of semantically-opposed stories is interesting. A well-executed version of this idea could form the basis for a quality metric.
2. The authors promise to open source their data and code.

**Weaknesses:**

This work has a number of major issues with execution quality, soundness, and presentation. Additionally, the work does not, in this reviewer's opinion, make substantial progress on an important problem.
1. Importance. This paper introduces 2 metrics of SAE quality (contrastive score and independence score). Neither of these stand out as being especially unique as possible SAE quality metrics (e.g. they don't capture usefulness for a downstream task). The provided motivation is eliminating the need for LLM-grading; however, (a) this seems more like a nice-to-have than an especially important property and (b) existing metrics like the sparse probing metric from SAEBench already satisfy this property. (For context, SAEBench comprised seven metrics.) So overall, even if well-executed, it's not clear that this work would be sufficiently important for acceptance in a top-tier venue.
2. The introduced metrics don't clearly relate to interpretability. The contrastive and independence scores measure basic properties of representational geometry—specifically, whether some dimension exhibits large variance between particular examples or relative to a dataset mean. However, the paper provides no justification for why these geometric properties should correspond to interpretability. In fact, we see the opposite: without an explicit sparsity penalty, denser SAEs score higher. Given this, it's likely that random up-projections, SAEs with random weights, or just using the base model's activations would score more highly on the metrics introduced here, despite being clearly less interpretable. More importantly, the authors never validate that latents with high contrastive or independence scores actually encode the intended semantic concepts.
4. Contrastive story generation seems dubious. Typically the goal of a contrastive pair (x, y) is for x and y to be as similar as possible except for differing with respect to a single concept of interest. That does not seem to be the case in the data introduced here. Looking at the example in table 4, the two stories both make clear reference to the concept with respect to which they're supposed to differ (computer) while varying in many other unrelated ways.
5. Various presentation issues. I will not detail these, because seeing them improved would not be sufficient to improve my score.

**Questions:**

1. How do the following baselines perform according to your metric? (a) The model's activations themselves (no SAE), (b) Random up-projections to a vector space of dimension d_SAE, (c) SAEs with random weights.
2. Requested ablation: Shuffle the stories in your data across constrastive pairs, then recompute your metric. Do the results look qualitatively different? If not, that suggests the contrastive pair structure in your data was not load-bearing.
3. Can you provide any evidence that the SAE latents with the largest contrastive or independence scores are semantically related to the relevant concepts? For example, in the "computer" example, do the top-scoring latents activate on computer-related tokens in other contexts?

---

### Official Review · Reviewer_ptSt · 2025-10-28

**Soundness:** 1
**Presentation:** 2
**Contribution:** 2
**Rating:** 2
**Confidence:** 4

**Summary:**

This paper contributes a new synthetic dataset based on contrastive prompts that can be used to evaluate SAEs. The paper also describes a new metric using this dataset to try to benchmark the interpretability of an SAE without relying on running LLM inference at test-time.

**Strengths:**

- The paper provides a new dataset that can be used as a benchmark for SAEs
- The paper is touching on a real area of need, which is better SAE benchmarks

**Weaknesses:**

- This benchmark appears to be a more convoluted and limited form of k-sparse probing, not auto-interp. It is not clear what value this provides as an SAE benchmark that k-sparse probing does not provide.
- The overall formula is unprincipled (summing 2 scores and subtracting sparsity with a coefficient just to make the numbers match SAEBench empirically). The sparsity component of this is particularly egregious; we want SAEs that are interpretable, we do not care about sparsity. We should not penalize SAEs that are interpretable for not being sparse. There is no inherent value to sparsity.
- The paper selects only a single SAE neuron to evaluate the difference between contrastive prompts. This assumes that the prompts differ in only a single dimension, and that this dimension corresponds to a LLM feature the SAE has captured. This makes this technique more limited than k-sparse probing. A `topk` instead of a `max` should be used, better aligning with k-sparse probing that SAEs are usually evaluated with.
- No error bars or stdev is provided for evaluations, and they appear very noisy, e.g. Figure 2
- The validity of the metric is not evaluated in and of itself, but only as a proxy to SAEBench's auto-interp score. SAEBench auto-interp is far from a perfect metric and should not be treated as such. The validity of the metrics in the paper should be evaluated in their own right.

**Questions:**

- Have you tried doing a standard k-sparse probing experiment with your dataset? This would be the more obvious way to evaluate an SAE using the dataset provided,
- You should explain what an SAE is mathematically in the paper. E.g. on line 128, it says "d is the dimensionality of the latent space", but this would be a lot clearer if had a background section explaining SAEs, and, presumably, d here means the dimension of the SAE latents, rather than the model latent space.
- On line 128, what is "min-max normalization"? In general it would help to write these operations in math notation.
- The metric described here is not being compared to SAEBench as a whole, which comprises a number of metrics, but only the auto-interp score. I feel like this is not stated as clearly as it could be in the paper, which repeatedly claims it is just comparing to "SAEBench" or "SAEBench interpretability". It sounds like this is specifically referring to SAEBench's auto-interpretability metric.
- On Line 297, regarding Matryoshka SAEs, it says "we exclude it from our evaluation because it lacks ground-truth annotations". What does this mean? What are "ground truth annotations"? Do you mean autointerp-scores? Why can't you run auto-interp yourself using SAEBench?
- Line 189 "pretrained sparse autoencoders (SAEs) publicly released by SAE-Lens", SAELens does not train their own SAEs, they only aggregate SAEs into a standard format for loading. You need to say what specific SAEs you are evaluating and cite the works that trained them.
- Line 197 "SAE-Bench interpretability scores are not publicly available". You can run SAEBench yourself, you don't need to rely on others to run it for you.

---

### Official Review · Reviewer_aqJr · 2025-10-31

**Soundness:** 2
**Presentation:** 3
**Contribution:** 2
**Rating:** 2
**Confidence:** 3

**Summary:**

Contrastive Evaluation Bench evaluates the ability of SAE latents to differentiate contrastive stories. The authors generate a dataset of 1000 contrastive paragraphs and measure two scores: the maximum difference of a latent on the contrastive pair (contrastive score) and the maximum absolute deviation from the mean activation vector (independence score). The authors highlight a 70% spearman correlation with the SAEBench Autointerp evaluation.

**Strengths:**

- Thorough investigation across multiple axes: SAE architectures, LLM layers, modules,
- Contribution of a contrastive dataset

**Weaknesses:**

1. Abstract: The claim that "most" SAE evaluation methods rely on LLMs is inaccurate, as only 1 out of 8 metrics in SAEBench use LLMs.

2. In Figure 1, the rarrow direction between LLM and SAE is possibly incorrect, potentially both arrows should point from LLM --> SAE.

3. The paper does not address how the evaluation generalizes or how large the dataset must be to capture all possible semantic contrasts. If an SAE performs well in discerning the contrastive pairs in the dataset, how confident are we this generalizes to other topics not contained in the dataset? A notion of error would be very helpful.

4. Sec 2.1: The filtering rules used for subject selection should be provided explicitly in the appendix.

5. The assumption that contrastive features must be mutually exclusive potentially breaks in the current setup. A feature might activate on related tokens in both contrasting stories (e.g., a "computer" feature could fire on "calculation" in the opposite description shown in Table 4).

7. The Figure 2 caption's claim that "CE-Bench interpretability scores show strong positive correlations with contrastive and independence scores" is contradicted by clear anticorrelations in some architectures (e.g., jumprelu).

8. Figure 1 is difficult to parse without reading Section 2 first; a figure illustrating the contrastive story pairs would be more suitable as the opening figure.

9. "introduce Spearman Correlation and Pearson Correlation" overstates the contribution.

**Questions:**

Suggestion: The paper should be framed as a standalone evaluation method rather than positioning it primarily as a cheaper replacement for LLM-based automated interpretability. A 70% Spearman correlation is insufficient to justify replacing existing methods solely for efficiency gains.

Could you elaborate on how better interpretability would follow from a higher independence score?

---

### Official Review · Reviewer_SLGE · 2025-11-01

**Soundness:** 1
**Presentation:** 1
**Contribution:** 1
**Rating:** 0
**Confidence:** 5

**Summary:**

This paper proposes a new benchmark for the interpretability of sparse auto-encoders. As part of this benchmark they generate a dataset of contrastive stories which is used as part of the benchmark. The proposed benchmark is cheaper to run than alternatives which use use LLMs. However, the motivation is poorly explained, and there is distinct lack of concrete examples of SAE features or dataset examples (paired with features) which makes it very hard to reason about whether this work is characterising interpretability of features in a meaningful way.

**Strengths:**

- Originality: The idea that SAE interpretability can be measured using paired dataset examples which differ in a single interpretable concept is interesting, and may be useful for identifying relevant concepts as represented by features in the SAE basis.

- (Weakly) Consistency with a previous metric is a basic sanity check which was good to include (though a more detailed comparison seems missing).

Other than that I see no strengths of note.

**Weaknesses:**

- The paper fails to engage with the object level data, relying entirely on summary metrics: The nature of what the benchmarks are trying to measure is completely obscured by the lack of examples. No specific features are mentioned. No max activating examples are shown. We don't see any examples of interpretability scores assigned by one metric / benchmark compared to another on specific examples. How can we reason about how interpretable SAEs / features are without looking at the underlying data? It's impossible to assess claims that one metric is a better measure of some underlying thing without looking at examples of the underlying thing. This issue cuts across the motivation for the paper (which is inadequately explained) and the standards of evidence (which seem poor).

- Standards of evidence seem poor in the paper. At various points, the authors gesture at already established ideas / results as being suggested  by their figures when the figures don't clearly make those points. For example, figure 2 - "Among all architectures, top-k and p-anneal consistently yield the highest interpretability, aligning closely with SAE-Bench ground truth". I didn't read that out of the graphs as all - the curves are all over the place, with p-anneal (red) and top-k (purple) not looking closer to each other than any of the other architectures.

**Questions:**

- Can you please more clearly illustrate how the contrastive stories represent concepts? Can you show examples of features which match stories and the scores assigned?
- Can you show the inadequacies or failures of previous interpretability metrics and indicate how your metric better captures what you believe to be the interpretability of different SAEs / their features?

---

### Note · Authors · 2026-01-22

I have read and agree with the venue's withdrawal policy on behalf of myself and my co-authors.